# Sustainable Stories: Managing Climate Change with Literature †

**Gesa Mackenthun**

North American Studies, Department of English and American Studies, University of Rostock, D-18055 Rostock, Germany; gesa.mackenthun@uni-rostock.de

† This essay is the result of a conversation with my fellow Scientists for Future collaborating on a bibliography on climate change. I promoted including literary texts as profound reflections on the present and future options. My colleagues, after remembering having themselves been influenced by literary texts—such as Ernest Callenbach's *Ecotopia*-finally allowed me to include Atwood's *Oryx & Crake*.

**Abstract:** Literary and cultural texts are essential in shaping emotional and intellectual dispositions toward the human potential for a sustainable transformation of society. Due to its appeal to the human imagination and human empathy, literature can enable readers for sophisticated understandings of social and ecological justice. An overabundance of catastrophic near future scenarios largely prevents imagining the necessary transition toward a socially responsible and ecologically mindful future as a non-violent and non-disastrous process. The paper argues that transition stories that narrate the rebuilding of the world in the midst of crisis are much better instruments in bringing about a human "mindshift" (Göpel) than disaster stories. Transition stories, among them the *Parable* novels by Octavia Butler and Kim Stanley Robinson's *The Ministry for the Future* (2020), offer feasible ideas about how to orchestrate economic and social change. The analysis of recent American, Canadian, British, and German near future novels—both adult and young adult fictions—sheds light on those aspects best suited for effecting behavioral change in recipients' minds: exemplary ecologically sustainable characters and actions, companion quests, cooperative communities, sources of epistemological innovation and spiritual resilience, and an ethics and aesthetics of repair.

**Keywords:** near future fiction; climate change; behavioral change; Margaret Atwood; Sita Brahmachari; Octavia Butler; Cherie Dimaline; Lukas Jüliger; Saci Lloyd; Sara Pennypacker; Kim Stanley Robinson

As William Blake noted long ago, the human imagination drives the world. Understanding the imagination is no longer a pastime or even a duty but a necessity, because increasingly, if we can imagine something, we'll be able to do it. (Margaret Atwood, *The Handmaid's Tale* [1] (p. 517).

I think hard times are coming when we will be wanting the voices of writers who can see alternatives to how we live now and can see through our fear-stricken society and its obsessive technologies to other ways of being, and even imagine some real grounds for hope. We live in capitalism. Its power seems inescapable. So did the divine right of kings. Any human power can be resisted and changed by human beings. Resistance and change often begin in art, and very often in our art—the art of words. (Ursula Le Guin, "Acceptance Speech" [2]).

When the story changes, everything changes. (Christiana Figueres and Tom Rivett-Carnac) (*The Future We Choose* [3] (p. 158).

## 1. Introduction: Limits of Perception

On 11 September 2020, California governor Gavin Newsom, addressing U.S. President Donald Trump in front of burning trees, stated that Trump's lowering of carbon emission standards was "beyond the pale of comprehension" as this privileging of the fossil fuel industry was one of the causes of the wildfires raging in California [4]. Trump repeated

his position that the fires were the result of imperfect forest clean-up, that the climate would cool off again, and that the scientists did not know what they were talking about [5]. Trump's climate change denialism, shared by millions of voters in the United States and elsewhere, is indicative of the limits of human perception which the authors of the Club of Rome report *Limits of Growth* (1972) had attested to the majority of Western citizens [6]. One impressive diagram in their fifty-year old classic on the limits of planetary resources shows the astoundingly small range of human perception which in the late 1960s hardly reached beyond the more tangible temporal and spatial range of direct experience (Figure 1).

**Figure 1 HUMAN PERSPECTIVES**

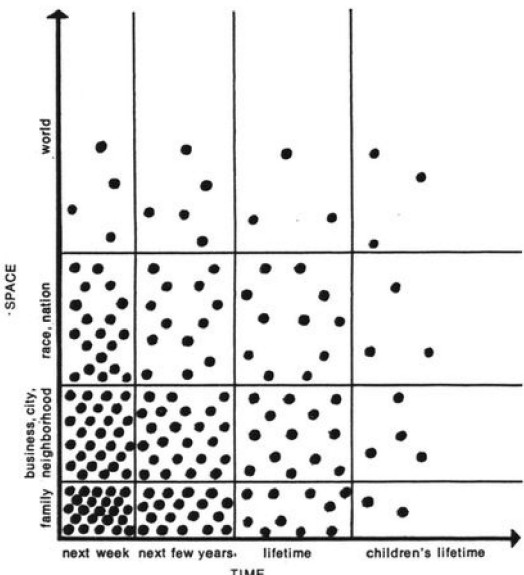

*Although the perspectives of the world's people vary in space and in time, every human concern falls somewhere on the space-time graph. The majority of the world's people are concerned with matters that affect only family or friends over a short period of time. Others look farther ahead in time or over a larger area—a city or a nation. Only a very few people have a global perspective that extends far into the future.*

**Figure 1.** "Human Perspectives." Donella H. and Dennis L. Meadows et al. *The Limits of Growth*.

　　　The incapacity to see a connection between climate change and the California fires derives from a hermeneutic lack of historical depth and spatial scope, as well as an inability to establish connections between causes and effects in a complex system. The fifty-year old diagram suggests a serious disjuncture between the finality of the Earth's capacity and humans' mental performance. In fact, while the world's economies became ever more entangled, complex, and ecologically devastating, the human power to imagine this complexity apparently did not grow proportionately outside the sphere of certain knowledge groups (such as scientists and intellectuals, the educated and interested public). While since the 1980s this imaginative gap may have diminished somewhat due to a trickle down of scientific knowledge, the massive rise of freely available web-produced fake information since the early 2000s, spread by disciples of antidemocratic and life-negating ideologies, has led to a new epistemic crisis. Next to robust government action for a speedy decarbonization of the economy by setting up the necessary technological infrastructure, the carbon-reduced rebuilding of society inevitably can only be accomplished with the consent of the majority of the population. This includes citizens' willingness to change elementary aspects of their lifestyle, citizens trusting that the knowledge produced by scientific experts is sound, and citizens being reassured that both experts and democratic governments are dedicated to truthfinding and the common weal. Under the impact of stories spreading doubt and fear, manufactured by a few unethically acting scientists and

conspiracy entrepreneurs [7], this trust has been severely eroded in the past decades. But just as stories are part of the problem, they can also be part of the solution.

In science fiction writer Kim Stanley Robinson's most recent novel, *The Ministry for the Future* (2020) [8], a narrative voice directly addresses the mentioned imagination deficit, here termed "universal cognitive disability." When a deadly heat wave with a few million human casualties strikes India at the beginning of the novel, most citizens of Western nations are able to wave away the problem because they are not immediately affected by the catastrophe. But even as another heat wave hits various states in the American South and Southwest, the magnitude of the danger is belittled in the U.S. north because something like this "couldn't happen to prosperous white people." Apparently, the voice wonders, people generally have "a very hard time imagining that catastrophe could happen to them, until it did. So, until the climate was actually killing them, people had a tendency to deny it could happen" [8] (p. 349).

This paper argues that the cognitive capacity of distinguishing fact from fiction, as well as grasping the long-term and long-distance connections involved in the phenomenon of climate change, crucially depends on hermeneutic powers trained by literature. However, literary texts and other cultural fictions also contribute to the imaginative repertoire of climate change denialism and pessimism. The frequent combination of denialism and the belief in secret conspiracies taking place behind the back of the general public suggests that at issue is less an incapacity of the imagination as such than a desire for a very particular kind of story: Ultimately, denialists crave for a modern version of an ancient crypto-religious fantasy of the "people" as victims of a devilish plan, combined with a desire for authoritarian rule and a lack of confidence in the competence of democratic and science-based governance. Most of the near future stories circulating in the popular cultural field possess little to no utopian potential; their moral messages are rather inadequate teachers of ecologically and socially sustainable behavior. The purpose here is to discuss a heterogeneous number of near future speculative fictions that depart from the anti-utopian disaster mode and bear the potential of helping their readers to imaginatively cope with the present ecological challenge.

As Martha Nussbaum has argued in *Poetic Justice*, literature, more specifically the novel, is formally equipped to "construct empathy and compassion in ways highly relevant to citizenship" [9] (p. 10). Nussbaum is predominantly thinking of a very particular kind of literature: realist British and American novels of the late nineteenth and early twentieth centuries. She argues that the novel genre, by virtue of producing an imaginative familiarity, even sympathy, with people and places remote from the readers' lived reality, has a great potential to train readers' capacity for empathy with humans outside of their immediate peer groups. As empathy and compassion are important preconditions of cosmopolitan justice, literature can educate the reader to develop as sense of social—and by extension ecological—justice even while causing entertainment and delight [9] (p. 6). Narrative's productivity, Erin James adds, derives from its ability to make readers immerse themselves in the lived realities of geographically and culturally distant humans [10] (p. 39). (Note: Like Suzanne Keen [11], I reject overgeneralizing claims that the reading of literature as such is a training program in cosmopolitan empathy. Yet, as I will argue, literary texts make different kinds of offers to readers, and some of these are more conducive to an ethic of ecological sustainability than others. The actual effects of any text are always the results of its interaction with specific readers.)

Climate change denying and anti-ecological ideologies are themselves inspired by a staggering number of imaginative disaster stories whose potential for generating empathetic response in the recipients is questionable, as their main effects are fearful delight and pessimism. (For a partial list of climate catastrophe novels and films, see Horn [12], p. 14.) Western culture is fatally absorbed in stories announcing the destruction of the world but lacking the socially productive features Nussbaum writes about. The apocalypse, whether ecological or more general in kind, has been announced and foretold so many times for millennia by the most powerful agents of world history that it today

amounts to a powerful collective imaginary shaping recipients' aesthetic taste and social attitude toward the future—and consequently also toward the present. This may in part be the result of the genre influence of SciFi and disaster fiction on climate fiction (CliFi) as well as climate *change* fiction. As Goodbody and John-Putra note, generic norms weigh particularly heavy on CliFi thrillers. Genre fiction is arguably a possible "resource" for thinking through "complex issues" like climate change [13] (p. 4). But due to its dependence on mass marketability, genre also imposes limits on aesthetic experiment and novelistic invention—including "ambivalent characters" and multi-dimensionality. The near-future disaster movie *Geostorm* [14], to pick a random but typical example, features a heroic male individual trying to prevent the end of the world by combating a deep-state conspiracy manipulating geoengineering devices. The thematic standard mix of these tales, consisting of malfunctioning high tech, evil-minded government agents about to destroy mankind, and masculine heroes who save the world from certain destruction, infiltrates pop culture consumers' perception of the real world with factory-made repetition. A particularly successful mode of near future narrative, building on the conviction that the world is doomed and will not be saved, simply because there are no convincing agents of a peaceful transition, is the climate thriller. Set in the near future, one of these (Dirk C. Fleck's *GO! Die Ökodiktatur*) [15] imagines a political takeover by an eco-dictatorship while another (Sven Böttcher's *Prophezeiung*) [16] invents a scientific-political clan who maliciously hides the facts about future climate catastrophes from the population. Other fictions represent scientists as greedy entrepreneurs (Ian McEwan, *Solar*, 2010) [17], tragic and vengeful figures traumatized by the disappearance of the natural phenomena they studied (Ilija Trojanow, *Eistau*, 2011) [18], or at best satirical figures such as the protagoist of T.C. Boyle's novelistic treatment of the Redwood Wars of the 1990s in *A Friend of the Earth* (2000) [19], in which a defunct elderly ecowarrior spends his old life as the keeper of a rich popstar's private zoo filled with extinction-doomed species. (For a critique of the CliFi conspiracy plot, see Andersen [20], chapter 4.) As in non-literary conspiracy theories, which these novels amplify and inspire, the conspirators' motivation is as shady as the means by which they obtained their democratically unchecked, quasi-totalitarian power. While scientists in the "real" world have been warning mankind about the consequences of the greenhouse effect for fifty years, the fictional scientists of these novels act out of egotistic motives by either covering up knowledge about the impending destruction of mankind or aiding eco-dictatorships. If their aim was to raise consciousness for the necessity of ecological change, most of these fictional stories failed. (Note: As Gabriele Dürbeck notes, Fleck's plea for an eco-dictatorship was widely rejected by his critics [21], (p. 253). Yet, scientists Oreskes and Conway likewise imagine ecological change to be brought about by authoritarian China as the democratic governments were too slow in dealing with the crisis. Imaginatively speaking from the future, presumably after the "great collapse" in 2093, they wonder if the Second People's Republic of China will voluntarily return to a democratic form of government [7] (p. 52). These speculations suggest that imagining the future environmentally is tantamount to imagining it politically. The process of political transformation from democratic governments to non-democratic regimes is rarely addressed in near future novels.)

It is hardly a coincidence that the toxic conspiracy stories produced out of a pessimistic, even defeatist disposition toward the world are concurrent with antidemocratic tendencies in governments and public discourse, whether online or physical. In addition, symptoms of "eco-anxiety" or "climateanxiety" are on the rise, as psychologists have been documenting for quite a few years [22–25]. "Every forest fire or starving polar bear is a jolt of stress," a reporter summarizes the situation in the *Guardian* [24]. Climateanxiety is beginning to replace the well-established and socially less dangerous end-of-the-world *Angstlust* which recipients draw from dystopian fictions. While dystopian Angstlust (anxiety delight) confirms its Western consumers in their middleclass complacency, eco- and climateanxiety begin to erode the foundations of society by lowering the cognitive and functional productivity of individuals [23]. Caroline Hickman of the Climate Psychology Alliance

told a *Guardian* reporter: "There are hundreds of people contacting us, looking for support. We've reached a level where organisations are asking for professional help to support their staff: civil servants, museums, universities. They're noticing massive increases in anxiety and concern" [24]. The Covid-19 pandemic is expected to aggravate the collective fear of the future.

## 2. The Pathology: Angstlust and Fictional Ecoterrorists

Asked in an interview with dragonfly.eco why he chose climate change as a topic in *The Lamentations of Zeno* (*Eistau*) [18], Ilija Trojanow pointed out that "good literature has always tackled the main issues of its time. Not to write about it would be weird, would mean succumbing to the blindness of an age that is pillaging the present and burdening the future." And he continues: "We have had an enormous amount of dystopian narratives in recent years, not only in literature but also in the movies, on the TV screen. We lean back, munch popcorn and delight in the apocalypse. That's pathological. To form a vibrant and dignified and truly humane future we need to imagine it first, we need utopian (or eutopian) ideas, concepts, narratives. We could do so much better, why not imagine it within a novel?" [26].

Some of the novels to be discussed here, like Trojanow's own, echo elements of the ecothriller and its anti-science ideology. Margaret Atwood's scientist figure Crake in *Oryx & Crake* (2003) [27] resembles other contemporary "mad" scientists in his pursuit of creating a humanoid species to inhabit the largely uninhabitable world cleansed from humans. He is aided in his fatal machinations by a social constellation in which a totalitarian form of corporate science reigns supreme, unchecked by political or non-parliamentary social institutions. Although Crake's terrorism as a GMO expert, manipulating nature according to his conviction that the world is better off without homo sapiens, differs from the sad vengeance of Trojanow's protagonist, both belong to the category of the scientific ecoterrorist, an outgrowth of a widespread suspicion toward the sciences in general. As other dystopian near future narratives, these are cautionary tales inviting readers to develop a critical attitude toward fatal developments in present societies. The cautionary function of climateanxiety and ecoterrorist stories will at best be a clarion call to action, at worst reinforce the conviction that the future is already determined and cannot be changed. This reflects a deep cultural anxiety that can be traced back to the apocalyptic master narrative, deprived of a utopian sense of post-apocalyptic survival. (Note: As John Hay points out, twenty-first-century postapocalyptic narratives, which he traces back to the Puritans, usually envision America as a wasteland inhabited by few well-prepared survivors [28] (p. 13). Not only has American reality been shaped in the face of that narrative of impending cathartic destruction but the story has "infected" the global culture industry. As Goodbody and John-Putra point out, disaster narratives with their (Western, Calvinistic) emphasis on guilt, punishment, and redemption sit ill with the open-endedness of climate change [13] (p. 11).) Climate anxiety and ecoterrorist narratives are unproductive in the sense that they have no empowering effect on readers; they do not imaginatively prepare them to cope with the present situation; and they train readers to make dangerous political choices. (Note: Frederick Buell gives a lengthy critique of the construction of environmental activists as ecoterrorists in anti-environmentalist discourse in the U.S. [29]) The antihumanist and apocalyptic mood that they endorse is amplified by bestsellers, written by academics, which present the end of humanity as the logical consequence of mankind's biologically engrained flaws, such as Yuval Harari's *Sapiens* (2011) [30] or Nathaniel Rich's *The Decade We Could Have Stopped Climate Change* (2019) [31]. But, as Naomi Klein and Elizabeth Povinelli point out, it is not "human nature" that is killing the climate but capitalism [32] (pp. 243–252); not humans in general but "a specific mode of human society, specific classes and races and regions of humans" [33] (p. 12). Violent catastrophism, writes Cory Doctorow (author of *The Walkaway*), is "a lazy trope of fiction." The fear at the heart of disaster fictions is based on the "belief in the inhumanity of our neighbors and, in particular, the inhumanity of neighbors who are poorer and browner than us." Doctorow refers to this

attitude as "elite panic" [34]. Elite panic and the fictions that both represent and reproduce it prevent acceptance of solutions suggested for the global recovery from environmental degradation. The fantasies they produce—of corrupt state power, of heroic self-sacrifice and white (masculine) victimization—aggravate conflicts rather than working toward solving them. This, Doctorow contends, is an "enormous impediment to a graceful recovery from a failure in our social and political and technological systems" [34]. Capitalism, with its bifurcated future narrative of economic bliss and impending apocalypse, has blotted out alternative stories. The "spirit" inspiring its "doomsterist" subplot (that dark partner of the Protestant ethic) has foreclosed the invention of alternative futures. (Note: "Doomsterism" is Frederick Buell's term [29].) I therefore share Gregers Andersen's position that disaster imaginaries are too reductive as a viable cultural response to climate change and that the "many dimensions of anthropogenic global warming call for a cultural analysis that takes other kinds of imaginaries into consideration as well" [20] (Introduction).

A critical reading of CliFi disaster stories may generate an understanding of the mental disposition that has assisted the "Capitalocene", or "Cthulhucene" (Haraway's term) [35], to drive the world to the brink of destruction. But the dystopian master disaster narrative, with its "tragic story with only one real actor, one real world-maker, the hero . . . the Man-making tale of the hunter on a quest to kill and bring back the terrible bounty" [35] (p. 118), has no cultural potential for shaping the global future in a positive way. This master narrative may "make" the present [12] (p. 24), but a present that remains locked in paralysis about the anticipated catastrophe. Disaster narratives' delightful end-of-the-world scenarios are the source of their success, which, as Eva Horn writes, seems paradoxical, especially as the late twentieth-century and early twenty-first century catastrophe tales, unlike the biblical tradition, do not imagine a new start [12] (p. 27). The recency of climate change disaster narratives—measured by the length of general knowledge of climate change—seems to be the result of widespread climate change denialism and the abovementioned hermeneutic blockage rather than the unrepresentability of climate change, as Horn argues [12] (p. 16). (Note: Horn [12] (pp. 16, 32, 24) refers to climate change as a negative event ("Negativereignis"), a slow-moving catastrophe "without event." Timothy Clark is likewise skeptical of the narrative form's capacity for representing climate change [36] (pp. 80–81). These claims were earlier voiced by Ursula Heise who, in *Sense of Place* (2008), dismissed climate fiction's power of representation, calling it an "exercise in second-hand nonexperience" [37] (p. 206). However, the climate change-related reality of the past three years with their droughts and terrible wildfires may offer reasons for revising this position. Next to the older iconic images of starving polar bears, mediatized images of dried-up agricultural fields, starved livestock, and burning forests are well suited to evoke the "events" causing such destruction. The carbon-emitting units causing these events are likewise representable. In short, there are no *narrative* limits to representing climate change, apart from the inherent mimetic limits of representation as such. My point is that there is an overabundance of a *certain kind of narrative* (disaster, doom, elite corruption/conspiracy), coupled with a lack of more sophisticated narratives adequate for representing the complexity of "dwelling" in a world ruled by climate change. As Gregers Andersen writes, "[i]t is precisely because we do not know how anthropogenic global warming will affect 'places and individuals' that climate fiction is such an exciting medium for reflection at this crucial point in human history" [20] (Introduction). Literature is the cultural laboratory for imagining the lived experience of climate change.) Because of the unsuitability of end-of-the-world disaster fictions for building a livable world to be inhabited by future generations, the focus here is on a different kind of story: speculative near future social-ecological transition narratives written in the context of climate catastrophe yet containing a faith in human cooperativeness and creative resourcefulness.

Referring to Margaret Atwood's predilection for the term "speculative fiction" over against "science fiction" with reference to her own writing, Chris Vials specifies that "speculative fiction relies on imagination and projections, but unlike science fiction, its plots and settings hue much more closely to empirically observable, social and technological

trends." This ranges speculative near future writing closer to literary realism [38] (p. 239; see also [1]). What distinguishes speculative future fiction from science fiction is that it is "self-consciously guided by an awareness of the social relationships of its own historical moment" [38] (p. 239) and imaginatively extends the analysis of the present into the near future. Near future transition stories, as here understood, paint a world in upheaval and disrepair but they focus on groups of individuals actively adapting to changing conditions, strengthened in their effort at survival by reciprocity and mutual aid. All novels to be discussed, moreover, represent the future in critical dystopian terms: They are cautionary tales rather than pessimistic announcements of global apocalypse. (Note: On critical dystopia as cautionary tales, see Moylan [39] (p. 184. Although this analysis is limited to novelistic texts, drama and poetry, too, have the potential for imagining, articulating and performing transition.) Although disaster does take place, it is rarely narrativized or dramatized, the cathartic spectacle of end-of-the-world scenarios is avoided. Rather, the diegetic focus lies on the power of individuals and small groups adapting to a ruined and changing world during and after environmental and social disasters. The cultural work, and the value, of these narratives rests in their potential to plant hope and confidence in their readers and to offer possible paths for survival in a world increasingly impacted by climate change and its effects. These stories respond to economist Maja Göpel's appeal to bring about a "great mindshift"—to create positive visions of the future without blindly imitating the shallow and ineffectual optimism of the capitalistic narrative of possessive individualism and fossil fuel-based progress [8]. Change, as Göpel correctly contends, begins in the realm of narratives.

The need for more realistic, critical yet still ecologically, socially and economically optimistic narratives is answered by writers from different cultural and national backgrounds such as Octavia Butler, Sita Brahmachari, Cherie Dimaline, Kim Stanley Robinson, and Saci Lloyd, and to some degree Margaret Atwood, Sara Pennypacker, and Lukas Jüliger. Some of these novels are written for adults, some for young adults. What unites them is their attempt to escape from the grip of (post-)apocalyptic disaster narratives and to enable their readers to cope with the challenges caused by species extinction and climate change by activating, in more or less realistic settings, ancient forms of individual and collective resilience. These social-ecological transition narratives have the potential to generate transformational energy for shaping the future by reminding their readers of well-tested survival strategies of the past.

## 3. Materials and Methods

*The Cure: Social-Ecological Transition Stories*

The disastrous reality of dying forests, environmental refugee movements, crop failures, and an increase of authoritarian and rightwing ideologies can no longer be denied. For a growing part of humanity, especially but not exclusively in the global south, dystopia is not merely a narrative but a daily experience: "Aboriginal people live in a dystopia every day," says Aboriginal writer Claire Coleman [40]. As the catastrophic consequences of climate change and ecocide are becoming more tangible, politically aware writers have begun to imagine socially and economically feasible future stories whose deployment of plot, character constellation, tropes, and voice likely has empowering effects on attentive readers. The most prescient of these writers, African American writer Octavia Butler, imagined not only an American West Coast on fire and in social upheaval but also a populist U.S. president promising to "make America great again"—in 1998 [41] (p. 24). (Note: Butler was one of the earliest novelists to recognize the severity of the greenhouse effect in the 1990s and to regard the climate change topic in conjunction with neocolonial trade policies like NAFTA [42] (pp. 83–100).) Others, especially authors of young adult (YA) fiction, appeal to the empowering comfort of ancient knowledge adapted to present needs, as well as to the solidarity between humans in emergency situations.

Social-ecological transition stories do respond to Trojanow's appeal—to imagine a "vibrant and dignified and truly humane future"—to the extent that they include an

aesthetics of repair and rebuilding which is often rooted in ancient cultural technologies of survival. Transition stories, I argue, form an alternative archive for imagining the near future. Though including references to the disastrous effects of climate change, ecocide, and their social consequences, these texts choose not to focus on spectacular violence but rather imagine the future as a period of painful but manageable transformation, achieved through companionship, adaptability, and enthusiastic labor. What, I want to know, is the novels' cultural work? What is their potential for facilitating social and behavioral change? How do they address readers affectively, how do they articulate—and contain—their anxiety of impending social crisis and climate disaster?

It is useful to distinguish between a thematic and a formal level. On a thematic level, social-ecological transition stories ideally narrativize an orientation to the common weal and toward material and intellectual subsistence. Literature can encourage sustainable behavior by representing exemplary, e.g., carbon-avoiding, actions of believable characters. Formally, literary texts' "sustainability" works through their structure of appeal—their ability to inspire readers to reflect on and imitate the fictional action. This requires narrative access to readers' imaginations which is usually best achieved through the deployment of exemplary characters (inviting identification) and deep cultural plot structures producing a positive resonance in readers.

Of course the contribution of individual literary works to a sustainable future is impossible to measure statistically. My discussion will rather focus on those aspects of the novels most likely to strike a chord with readers. Next to their representation of disaster, socioeconomic conditions, and ecological transformation, these include five interrelated aspects that occur frequently in the selected novels: companion quests, cooperative communities, subsistence economy, intellectual resilience through knowledge transmission, and an aesthetics of repair.

## 4. Disaster in Social-Ecological Transition Novels

Unavoidably, all novels do include, without elaborating on, ecological disaster, usually in league with social crisis and disintegration. Octavia Butler's *Parable of the Sower* (1993) [43], an early novel featuring the effects of climate change, depicts the collapse of a violently neoliberal social order, coterminous with the depletion of natural resources (water) and the eruption of deadly wildfires in California. A socially mixed group of people, who lost their homes either through human violence or fire, assemble on their way north where they are able to start a humble farming community. As the sequel, *Parable of the Talents* (1998) [41], explains, the disaster is very much the result of political incompetence and the rule of corporate power. Taylor Bankole, one of the founders of the rural survival community, informs his companions that the period between 2015 and 2030 was referred to as "The Pox," a coincidence of "climatic, economic, and sociological crises" [41] (p. 8). However, in his analysis this conjunction began much earlier:

"It would be more honest to say that the Pox was caused by our own refusal to deal with obvious problems in those areas. We caused the problems: then we sat and watched as they grew into crises. . . . I have watched education become more a privilege of the rich than the basic necessity that it must be if civilized society is to survive. I have watched as convenience, profit, and inertia excused greater and more dangerous environmental degradation. I have watched poverty, hunger, and disease become inevitable for more and more people" [41] (p. 8).

Bankole here expresses the deep implication of individual citizens like himself in allowing society to deteriorate. "Butler does not present crisis as a rush toward apocalypse but as a *dwelling place*," writes Frederick Buell, "as a context in which people struggle against growing odds to be able to live some sort of normal life" [41] (p. 314; emphasis added).

In her novels *Oryx & Crake* (2003) and *The Year of the Flood* (2009), Margaret Atwood, too, describes disaster—a man-made pandemic as well as man-made climate catastrophe causing the extermination of most humans on earth—as what Buell refers to as "dwelling" in a place uncannily familiar to contemporary readers [29]. The damaged storyworld is the

result of a slow development toward a society ruled by corporate power in conjunction with an excessive popular culture of hedonism and violence. While *Oryx & Crake* narrates events before and after the cataclysm, the plot of *Year* includes narrative glimpses at the dreadful effects of the "Waterless Flood" (as the anticipated catastrophe is called) but concentrates on the few survivors, such as the religious cult God's Gardeners [44].

The YA graphic novel *Unfollow* (2020) [45] by Lukas Jüliger, too, ends with the destruction of mankind, masterminded through a geological power incarnate in the enigmatic protagonist Earthboi who manipulates the young generation via digital media to commit collective mass suicide. The catastrophe is brought about by Earth itself, here presented as agent and narrator, which defends itself against, or takes vengeance of, mankind. Friday kids and ecologically minded people are its deadly instruments without independent willpower. Jüliger deploys an antihumanist view reminiscent of Harari's *Sapiens* and other social Darwinist narratives of "human nature."

From Indigenous and African American perspectives, disaster has been a reality for centuries. Colonialism continues to deprive aboriginal people of their land, cultural sovereignty, and self-representation. The action of *The Marrow Thieves* (2017) [46] by Canadian Métis writer Cherie Dimaline takes place in a world affected by climate change-related events like rising sea levels and a massive reduction of the world population as a result of floods and loss of land. The remaining humans in this slightly fantastic YA novel are beginning to lose their ability to dream, going insane. After discovering that Indigenous people have retained their power to dream, the rulers of the land join up with churches and scientists who transform former residential schools into laboratories where Indigenous peoples' bone marrow, seat of their dreams, is extracted in order to be implanted in the bodies of white people. These terrible events are narrated by Miigwans, the elderly leader of a group of Indigenous refugees who join up with one another on their way north [46] (pp. 100–101). The climate disaster is merely the trigger of the new social condition which basically repeats colonial societies' intellectual "harvesting" (i.e., appropriating while destroying) Indigenous peoples' cultural knowledge.

In *Where the River Runs Gold* (2019) [47], the British postcolonial writer Sita Brahmachari invents a similar conjunction: Most of the geographically unspecific post-climate-disaster world has been destroyed by a gigantic storm, leading to a political takeover by a small elite which now enslaves the majority of the impoverished population to work in its gated gardens. While the earth is broken, as in the other novels, some hidden pockets of intact nature have survived. It is surprising that the four novels for young adults are hardly less graphic about the ecologically and socially catastrophic effects of the Western capitalist lifestyle than the "adult" novels.

Saci Lloyd's YA novel *Carbon Diaries 2015* (2008) [48] provides a quotidian perspective on a similar socioeconomic transition as Robinson's whose representation Robinson avoids. In the teenager Laura Brown's London, daily consumption is severely restricted by carbon cards whose number of points suffice merely for the social life of hermits. In the midst of climate crisis, Laura tries to cope both with the usual problems of an adolescent (parents, love interests) and the daily carbon card disaster. Meanwhile the flood sweeps over London, without causing too much human loss due to an ancient dyke. The future remains ominous.

Stashed away into Miigwan's story lesson in *Marrow Thieves*, wrapped into fairy-tale comfort by Brahmachari in *River Runs Gold*, and subject to liberating laughter in *Carbon Diaries*, man-made ecocide and social devastation are presented much more painfully in Jüliger's *Unfollow*—reinforced by rather scary graphics. Political players in all of these novels are no bulwark against disaster but a dark and anonymous power leaving little to no room for democratic agency. Especially Jüliger's novel favors a fatalistic, antihumanist concept of human nature, testing the moral limits of YA fiction.

Kim Stanley Robinson's *The Ministry for the Future* (2020) embeds these darker aspects of climate change-related catastrophe in an optimistic account of the near future. Here the economic transformation is initiated by global political agencies. Disaster takes the form of global heat waves with several million deaths. Transition happens on a global

scale, with various attempts at (mostly ineffective) geoengineering competing with the top-down establishment of a carbon-free economy. Human initiative takes two main routes, both of which are represented in a rather abstract way: state enforcement and terrorism, with the United Nations unit Ministry for the Future secretly condoning, perhaps even financing, terrorist activities. Robinson avoids the humanist approach evoked by Trojanow and Doctorow: We learn little about the intrinsic motivation of his two round characters, Mary Murphy (the head of the Ministry) and the humanitarian aid worker Frank May. The transition to an ecologically sustainable future is mostly a bureaucratic process orchestrated from above.

## 5. Socioeconomic Relations

Octavia Butler's United States of the 2020s and 2030s is ruled by a Christian- fundamentalist president modeled on right-wing evangelical figures of the 1990s like Newt Gingrich. President Andrew Steele Jarret comes to power during the social and ecological turmoil of the 2020s and starts resource wars against Canada and Alaska which secedes from the United States. As small local businesses start developing and things start to improve, Jarret establishes an authoritarian regime, coaxing enough citizens into voting for him in the hope of receiving stability and order in return [41] (p. 234). Autonomous communities are razed, their inhabitants incarcerated and their children abducted to be reeducated. Butler's novels paint an eerie picture of the United States as a country lapsing into corporate totalitarianism in league with right-wing fundamentalism. But in spite of the abundance of dystopian elements, her novels convey a utopian sense of the possibility of social change and repair, to emerge primarily from disfranchised social groups. Butler's protagonist Olamina's flight from her gated community in the Los Angeles metropolitan area is directly triggered by the violent attack on the community but also the result of the slow transformation of urban housing effected by real estate corporations who are enticing people to move into their compounds, giving up their freedom in favor of security. Together with other refugees she meets on her escape route to the north, Olamina builds up a rural community on a low material level which, lacking state protection, has to defend itself from omnipresent looters by force of arms. While the U.S. president raves about making America great again, people are rebuilding society from below.

In *Year of the Flood*, Atwood, like Butler, makes an investment into describing the process of economic rebuilding, exemplified by the rooftop horticulture of the God's Gardeners called Edencliff. But she treats her community with much more detachment. Her protagonist Toby, who joins the garden commune after being abducted by its members (and thus saved from a deadly fate as a sex worker), is unconvinced of the spirituality and reluctant about the garden work practiced by the group and its guru, Adam One. Social hegemony outside of Edencliff is enforced by an invisible corporate power with its own paramilitary security force. There is no sign of any government. The economy is dominated by the biochemical industry whose fanciful genetic splicings have led to the production of new hybrid animal species as well as the entertainment industry specializing in sex and violence. Next to the peaceful and vulnerable Gardeners, there exist various militant resistance groups. The usurpation of political power, however, is brought about not by human agency but by the Waterless Flood, a deadly multicatastrophe erasing most of mankind.

The other novels are less outspoken about the political and economic order but in all cases, democracy has gone down the drain. With the natural and social worlds in a state of devastation, authors imagine how common people cope with the situation of losing food security and social protection and being directly exposed to natural catastrophes like floods, fires, and pandemics. The intangibility of political rulers is a common feature of the YA novels. Society is run, and abused, by a neotribal private oligarchy in *River Runs Gold*. Dimaline in *Marrow Thieves* imagines a failing neocolonial regime desperately fighting for survival by forcefully harvesting the minds of colonized people. *Unfollow* makes no reference to any state power but, like *River Runs Gold*, illustrates the global dimension of

local events. *River Runs Gold* ends with the demise of the regime and its replacement by an elected government committed to global justice. *Unfollow*, conversely, envisions the end of mankind, brought about by intelligently manipulated individuals and by a vengeful Earth through its medium, the false messiah Earthboi. But like Yuval Harari and Nathaniel Rich, Jüliger does not distinguish between responsible elites and poor inhabitants of the global south with a comparatively small climate footprint. Of all novels discussed here, it is the most pessimistic one.

The two novels most outspoken about socioeconomic relations are Saci Lloyd's *Carbon Diaries*, which imagines a surveillance-based state bureaucracy that has imposed strict economic control on the population, and Robinson's *Ministry for the Future*, which is less articulate than Lloyd about the human reaction to state authoritarianism. In *Carbon Diaries*, the adolescents, suffering from the limitations imposed on a satisfactory life by the government's carbon cards, wind their way through the crisis with wit and dark humor. *Ministry* offers the most explicit socioeconomic vision for preventing the catastrophe into which the world is sliding in the 2020s. Its novelistic plot intersects with ambitious, cutting-edge theoretical reflections. Robinson imagines relief from climate change-related disaster as a consequence of concerted action of global state powers. His protagonist Mary Murphy, former Irish foreign minister and now president of the Ministry for the Future, is able to convince the central banks of the most powerful states to introduce a new currency (carbon coins) which binds economic success to a product's effectiveness in lowering the global emission of carbon dioxide. While preventing tax evasion by introducing blockchain currency, the new economy massively encourages ecological products and activities aimed at carbon reduction, such as reforestation, agroforestry, and rewilding. The official strategy of the global institution is aided by its clandestine branch which operates in league with terroristic practices against the world's ecologically most dangerous technologies (such as airplanes and container ships), corporations and individuals. This mixture of top-down politics and bottom-up terrorism eventually leads to a lowering of Earth's carbon footprint.

Robinson's novel, set between c. 2030 and 2048, presents a vision of the near future in accordance with the (presently waning) belief that the worst effects of climate change (i.e., a temperature rise of more than 2 degrees Celsius) could still be prevented if the signatories of the 2015 Paris Climate Agreement met their promises. In the novel, the global catastrophe is prevented by an economic elite accepting the truth of scientific predictions—such as the elite which regularly follows Klaus Schwab's invitation to Davos to deliberate on the global predicament—and a group of terrorists killing off members of that elite and making carbon-intensive travel unsafe. In fact, in one of the chapters one such terrorist group takes the Davos community itself hostage for a few days and subjects them to a radical education program [49] (pp. 159–164). The tycoons seem little impressed. (Note: In their technoeconomic real-life dystopia *The Great Reset*, written concurrently with *Ministry for the Future*, Klaus Schwab and Thierry Malleret propose a post-pandemic economic restart that includes an active regard for climate change-related requirements by limiting environmental degradation and air pollution while enabling biodiversity [50] (p. 206). They reimagine capitalism as a mix of stakeholder impact, "artificial intelligence" (read mass surveillance and automated decision making), and decarbonized lifestyles. Just how to organize this transition culturally—how to mitigate the expected economic and emotional shocks for the common citizens—is not part of their reflections.)

Robinson's "eco-socialist utopia" [51] implies that change will not be brought about by intrinsically motivated citizens and consumers. He does not employ the formal possibilities of literature, e.g., in dramatizing such individual conversions. Even his two round characters, Mary the United Nations bureaucrat and her initial antagonist Frank May, lack a realistic novel's psychological interiority. Instead, the novel consists of a polyvocal concert—sometimes chorus—of impersonal voices, including a Syrian refugee woman stuck with her family in Switzerland where she ultimately gains residency, and a team of engineers pumping water from the bottom of Antarctica to slow down the melting of the ice shield. Other scenes describe conferences of scientists and agricultural practitioners

exchanging their experiences. It ends with the good news about the global lowering of carbon emissions and with Mary, having reached retirement, joining a friend in his airship on a Jules Verne-like grand tour of the planet, witnessing the world's nascent transformation from fossil fuel to a more sustainable economy.

*Ministry* clearly delineates how the transition is orchestrated. But its success depends on a dark gap: The novel does not dramatize the transformation on an individual level. There is no scene showing citizens' difficulties in coping with the expected consequences of economic change which will interfere with the most basic daily practices, from meat avoidance to carbon-free mobility. There is only one scene that addresses the human response. The inhabitants of a Montana village are told that they will have to give up their homes because their area will be part of a rewilding program. These evictions, they learn, "were happening all over the upper Midwest, all over the West, the South, New England, the Great Lakes. Everywhere on Earth, we were told. You could buy an entire Spanish village for a thousand euros, we were told. Central Spain, central Poland, lots of Eastern Europe, eastern Portugal, lots of Russia" [49] (p. 438). In the real world, the private purchase of a Spanish, Portuguese, or Polish village primarily would seem to fulfill neo-aristocratic dreams of a sizeable hunting estate rather than contributing to decarbonization. This detail makes the depopulation of the Montana village reminiscent of the historical enclosure policy by which the land was originally grabbed by big landowners. But Robinson's assessment of the psychological effects of the buy-out of the villagers, as part of the "Half Earth plan" transforming rural areas back into wilderness, is realistic: "All these sad little towns," the narrative voice reflects, "the backbone of rural civilization, tossed into the trash bin of history. What a sad moment for humanity to come to." "Like a hospice preacher," their "facilitator" tells the crying townsfolk that someday in the future, "when the world's population drifted back down to a sane level, people would move back out of the cities into the countryside, and the villages would come back. Animals would walk right down the streets. They do here already! Someone shouted. Yes, and it will happen again, she replied. People who like knowing their teachers, their repair people, their store clerks and so on. The mayor. Everyone in your town. All that is too basic to go away for a long time. But now it's some kind of emergency" [49] (p. 440).

The scene encapsulates the brutal consequences of the urban-rural divide so deeply engrained in Western thought. It considers the persistence of ecologically restructured rural human settlements to be a handicap to decarbonization, suggesting instead their complete dissolution. The plan confirms the widespread absence in rationalist thought of a human sense of place—a love of place—which is here noted ("Everybody cried") but then sacrificed for an abstract idea of tabula rasa and a new beginning. It denies the possibility of humans living in peaceful cohabitation with their "wild" non-human fellow creatures, although humans have been doing this for thousands of years. Field scientists and park rangers will replace the rural town citizens [49] (p. 439) who are apparently unfit for becoming stewards of their homeland. The chapter, while reiterating the American myth of a vacant wilderness which caused historical forced relocations, reinforces America's national mythology of geographical mobility, only seemingly alleviated by the vague hope that some later generation might be allowed to move back to the countryside. As in a Malthusian experiment, populations are to be moved around without too much concern for their emotional wellbeing. The "local indigenous human populations," who are to act as game keepers of this apartheid world, are explicitly included in the migrating animal populations moving around in the wilderness corridors [49] (p. 556)—reiterating the dehumanizing discourse of five hundred years of colonization, oblivious to the changes colonialism has effected on Indigenous lifeways.

The fate of the Montana village forms an uneasy counterpoint to the novel's emphasis on elite travel represented by Mary, who can afford Atlantic crossings on solar-powered high-tech clippers. Thus, the novel's magical narrative of a successful transition depends on significant gaps, e.g., its avoidance of a realistic treatment of the question of how the townspeople of the globe will react to the necessary measures to be taken to enforce

decarbonization. The ubiquitous climate-denialism of right-wing populist movements, from the gilets jaunes and the German AFD to the 7,4 million Trump voters, suggests that the political inequality and technological innovations which Robinson's novel portrays would meet with massive protests. The fictional Montana town's emotional reaction to dispossession is marvelously mild in comparison with the real-life resistance to be observed against much less incisive interventions into peoples' lives.

While the novel's Montanans, compared to the real United States, are all too peaceful and resigned to their fate, any representation of the emotional world of ecologically minded citizens, who mostly appear as efficient but anonymous agriculturalists, is likewise avoided. As Anthony Skew suggests, it is probably no coincidence that the activities of the novel's ecoterrorists are not dramatized either. Skew is "hankering for Robinson to write a book about political violence. Hell, write a sequel to this book that tells the story of the Ministry's black ops wing—show us what it means to blow up aircraft, assassinate the heads of corporations and steal the wealth of billionaires. Robinson's eco-socialist utopia is incomplete because he hand waves all that away—even while acknowledging that it's necessary [in the world of the book] to get the outcome ultimately arrived at" [51].

The following sections will look at some of the novels' literary devices that characterize their sustainability as exemplary transition stories.

## 6. Companion Quests

Octavia Butler's *Parable of the Sower*, as well as some of the YA novels, choose the most ancient plot form for expressing their characters' coping strategies in a world shaken by social disintegration, violence, and existential threat. Historically known since antiquity (e.g., Homer's *Odyssey*), the middle ages (e.g., the *Canterbury Tales*), and the beginnings of the modern novel (with its central theme of the journey), the quest plot describes an individual or collective journey caused by social disorder and aimed at repair through the agency of forces external to the community's common radius of experience. This literary form is well suited for describing the concerns of (groups of) people heading toward the unknown. It is therefore also a frequent plot form in coming-of-age stories.

*Marrow Thieves* and *River Runs Gold* feature unhoused protagonists whose resilience depends on other beings whom they fortuitously encounter during their escape from colonial detention camps. The human groups in *Parable of the Sower*, *Marrow Thieves*, and *River Runs Gold* team up during their flights from disaster. Though led by strong individuals, the questers deployed in these novels depend on mutual aid. Such plots are frequent in folktales (e.g., the "Bremen Town Musicians") and in modern classics such as Frank Baum's *The Wizard of Oz*, where a group of homeless and disabled figures (humans and more-than-humans) band together and by mutual aid master many challenges [52]. In Salman Rushdie's allegorical YA novel *Haroun and the Sea of Stories* [53], a similar motley group defeats an evil force which has stopped up the Ocean of Stories.

Essential to companion quests is the effective symbiosis of different individual talents. As in many childrens' books where the smallest and most insignificant creatures are often the most resourceful and resilient ones, social-ecological transition stories show how individual vulnerability is overcome with the help of strangers—and how strangers become friends as a result. (Experts on children's literature point out this important plot aspect (see, e.g., [54] (p. 128).) In *Sower*, collective identity of this kind is shaped during the trip through a hostile territory and the pursuit of rebuilding the world and reinstalling order. Given Olamina's strong sense of spiritual leadership and renewal, it is perhaps no coincidence that the founding group of the Acorn community includes twelve members [55] (p. 197). Like Jesus and his disciples, Butler's motley group lives in a world full of violence and hypocrisy. As Lisbeth Gant-Britton argues, "readers can glimpse utopian flashes against the strong pull of the dystopian sociocultural, economic, and political downturn that the Parable novels forecast" [55] (p. 190). These non-conformist collectivities signal that "the future does not necessarily mean progress unless a transformed and much more morally engaged citizenry makes it so." Featuring such communities, Butler's narratives "prod readers to

engage the often painful present and imagine a more moral, egalitarian, and productive relation to the future" [55] (p. 186). Their sustainability rests on combining one of the oldest narrative forms with a sociologically reflected "moral enterprise" [55] (p. 187), with tangible results in the real world: numerous activist groups confess having been inspired by Earthseed, both within and outside of Butler's novels [55] (p. 187). (Note: It's the motley composition of these groups which distinguishes it most from the sociologically more homogeneous travelers of the American road novel, from Cormack McCarthy's *The Road* (2006) [56] to Clara Hume's *Back to the Garden* (2018) [57], which fictionally resemanticize the American myth of violent westward expansion.) Butler intensifies the moral message of her human constellations by adding her protagonist's special gift called "sharing," which causes her physical pain when observing hurt in other beings nearby. With this supernatural device Butler expresses the centrality of the human capacity for empathy in the successful establishment and maintenance of social groups.

Though the plot structure of Jüliger's *Unfollow* is partially that of a companion quest as well (Earthboi recruits his human followers, turning them into instruments of his power), the appeal of this form is disturbed by the distancing choral narrative voice, the voice of non-human nature itself. The narrative agency is ultimately, and disturbingly in a YA novel, closer to Puritanism's angry god taking revenge on humanity. Like Crake in Atwood's novel, Earthboi is ultimately a lonely avenger and destroyer, as if taken from a Jacobean revenge tragedy.

## 7. Subsistence Economy

The novels' deployment of gardens reflects the global rise of initiatives for rebuilding environment-friendly forms of agriculture, frequently in tandem with Indigenous and Blackstruggles for environmental stewardship [58]. The fictional economies have undergone a return to subsistence food production. In *Unfollow*, Earthboi teaches the other children how to sow, plant, and reap. The God's Gardeners practice (urban) gardening in Atwood's *Year*, as do the Acorn settlers in Butler's novels. In the satirical *Carbon Diaries*, the garden sequences produce comic relief as Laura's father, by profession a college teacher, starts a garden in their London home backyard, complete with chickens and a pig called Larkin (after the poet, Philip Larkin). But Lloyd's satire also epitomizes all of the novels' reluctance about investing more than superficial literary energy in describing or imagining human interactions with plants and farm animals.

Octavia Butler is most articulate about the benefits of garden work, although the protagonist prefers the *metaphorical* fertility of her spiritual mission, Earthseed. In *Talents* a vegetable garden is the setting for Olamina's attempt to win the confidence of a young, emotionally challenged woman she meets during one of her journeys. Len savagely rejects Olamina's Earthseed faith, calling it "a lot of simplistic nonsense" [41] (p. 348). However, while weeding a veggie plot for another woman who had moved from the Bay Area to the surrounding hills, keeping a vegetable garden and some animals on rented land [41] (p. 352), Len overcomes her initial antipathy to both the "dirty" work and Olamina's spiritual message [41] (p. 349). Emotional healing and intergenerational dialogue are facilitated by cooperative farm labor.

The reasons for the reluctance to spell out the details of veggie gardening must in part be sought in the American colonial past. Historically, horticulture is often associated with hard physical labor and "dirt," a low social position, and states of violent subjection. *Where the River Runs Gold*, whose child characters are exploited as slave workers in the pollination tunnels and fields of a state "school," is a reminder of the abuse of Indigenous children as cheap labor force on the farms of colonial boarding and residential schools, many of them run by Christian churches. The colonial slave-based economy in the Americas of course rested on the violent exploitation of African slaves and other socially degraded humans as agricultural workers. While members of the colonial elite, especially in England, enjoyed their flower gardens, food production often rested (and still often rests) in the hands of underprivileged domestic and imported dependent laborers. These historical traumata

may explain the general aversion to positive representations of agricultural work in colonial and postcolonial literature. Kim Stanley Robinson's novel tries to escape this burden by presenting India as the leading country with respect to organic agriculture. In one chapter, a couple starts a carbon-sequestering garden, having their no-till agricultural work richly rewarded with carbon currency [49] (pp. 399–401). Another chapter has an assortment of global agricultural initiatives proudly exchange their experiences, especially in the area of agroforestry, at an international conference [49] (pp. 425–28). However, as the other texts, *Ministry for the Future* is reluctant in spelling out the details of subsistence gardening and in presenting it in an aesthetically pleasing and intellectually rewarding way.

## 8. Intellectual Resilience through Education

The acquisition of knowledge is a central theme, especially in the YA novels. In Atwood's novels, higher education is strictly separated into the well-financed corporatized colleges dedicated to the hard sciences and located in the compounds of the elite and the underfunded humanities located in the so-called "pleeblands." It is a vicious comment on the destiny of the humanities in a profit-oriented dehumanized society. Meanwhile, the God's Gardeners have given up writing as a technology; knowledge is transmitted orally during Adam One's religiously inspired lectures and songs.

The educational system in Butler's America ruled by Christian-fundamentalist president Jarret has ended public schools. Like housing, education is conducted by companies who "offered security, employment, and education. That was all very well but the company that educated you owned you until you paid off the debt you owed them. You were an indentured person, and if they couldn't use you themselves, they could trade you off to another division of the company. . . . You, like your education, became a commodity to be bought or sold" [41] (p. 351).

Olamina's daughter, taken from her during a government attack on her farming community in northern California, is a victim of the ideology of colonial reeducation and child abuse by her foster father [41] (pp. 252, 236). Here as elsewhere, Butler's speculation about the near future is indebted to her knowledge of the slavery past and its selective education of non-white children to become part of the national workforce. Butler emphasizes the importance of education as a precondition for a sustainable world. Her characters collect knowledge in all forms, from books to digital. One of their first operations after arriving at the remote farm to escape the general turmoil is to establish a school.

The other two writers of color likewise fuse the topic of school education with memories of colonial education. This is especially striking in Dimaline's *Marrow Thieves* in which the former residential schools—historically the traumatizing sites of massive colonial deculturation and reeducation—are transformed into laboratories for depriving Indigenous people of their power of dreaming, located in their bone marrow. Allegorically, this is of course precisely what the colonial reeducation program did and intended to do with generations of Indigenous children. Especially in Canadian writing, the residential school has become a literary topos expressing the cultural genocide carried out as part of the assimilation policies of the twentieth century. In a central scene in *Marrow Thieves*, the elderly, apparently senile woman Minerva is caught by the marrow recruiters, brought to a residential school laboratory and hooked to the machine to harvest her marrow. Unexpectedly, her humming and singing, aided by tribal members singing and drumming in support outside the compound, ruins the machinery. Although she dies, the recruiters are not able to appropriate her dreams. Brahmachari's *The River Runs Gold* features a residential school-cum fruit orchard where children are forced to work as artificial pollinators, bees having become extinct. (During her flight, the protagonist Shifa discovers bee survivors in a hidden valley which is celebrated as a sign of hope.) The school administration in *Carbon Diaries*'s authoritarian system polices the youth's education and punishes non-conformers by sending them to the Carbon Offenders Recovery Program. Finally, in Jüliger's *Unfollow* Earthboi is sent to a correctional institution because of his strange affinity for dead crea-

tures. He there meets his human followers who organize his escape and disseminate his message via the internet, unaware of the fact that they are used by a non-human, geological intelligence set on destroying mankind.

In a transitional world shaken by ecological and social upheaval, many texts imagine epistemic transmission to take place orally, through storytelling and song. Butler's teachers and Atwood's God's Gardeners have regular sessions for sharing knowledge; Dimaline's small Anishnaabe refugee community is held together by Miigwans' storytelling sessions used for preserving important knowledge of recent historical events, as well as for sharing ancient stories providing emotional strength and comfort. A character unfamiliar with the stories of residential schools falls victim to the bodysnatchers. Knowledge and education are essential for survival.

## 9. Spirituality as Source of Power

With their emphasis on epistemic change, supported by an unorthodox, pantheistic belief system, Butler's novels have left an imprint on the real world in that they inspired the formation of "intentional communities" in the United States [55]. It is the spirituality expressed in the Earthseed creed, rather than the horticultural practice of the Acorn community, which was adopted by these groups. It may seem strange to secular readers that the American writers imagine new belief systems as a necessary component of their near future societies, as even one character states in the otherwise very secular *Ministry for the Future* [49] (p. 546). Olamina, the protagonist of Butler's two novels, is guided by her eschatological belief system derived from the teachings of her father, a university professor and Baptist preacher. Its central dogma is the belief in "change" ("God is change"), a radical inversion of the Puritanical dogma spelled out by Jonathan Edwards in his sermon "Sinners in the Hands of an Angry God" (1741), that God "'abhors you'" [41] (p. 61). Olamina is convinced that the future lies in the settlement of distant planets and that Earthseed will "take root among the stars" [41] (p. 44). (Note: This recalls, but also essentially differs from, Ursula LeGuin's figure Odo, founder of an anarchistic society on the desert planet Anarres, in *The Dispossessed* (1974) [59]. The intention to terraform a new planet, too, resonates with Le Guin's novel in which a group of radical discontents leaves the planet Urras in order to establish an anarchistic society. Just like LeGuin, Butler's dialogic and contrapuntal structure invites her readers to critically assess the portrayed positions.) Olamina's belief that the "destiny" will be fulfilled in outer space, echoing neo-slavery fiction's conviction that there is no escape from slavery on earth, arises from her desire to rebuild the world as a tabula rasa, shaking off the historical burden of colonialism. She imagines Earthseed as living "in partnership with one another in small communities, working out a sustainable partnership with our environments [,] treating education and adaptability as the absolute essentials that they are" [41] (p. 343). It will require an "essentially human" form of religion to counter the "folksy stories" of her ideological antagonist President Jarret, who promises his adherents "the moon and stars" [41] (p. 344). But she has to concede that her project failed for the time being while "'Jarret's Crusaders [are] still running loose'" [41] (p. 344). At the end of *Talents*, she begins rebuilding her community with the financial support of wealthy humanistically minded sponsors. Her more educated companions reject Olamina's creed but take faith in the social experiment of cooperative work. Butler's interstellar brave new world is not narrativized; her novels rather point out strategies for coping on a damaged planet, for, in Haraway's terms [35], "staying with the trouble" in the midst of a continuing crisis that will not recede very soon. It is probably this realistic assessment of the precarious present—of the necessity of "making kin" in a ruined world—that most convinced her readers.

The God's Gardeners in Atwood's *Year* resembles Earthseed in maintaining food sovereignty in a morally and ecologically ruined world. Under the guidance of Adam One, the Gardeners practice a pantheistic companionship in Edencliff Rooftop Garden. In regular meetings, sermons, and songs, Adam One instructs his followers—a similar social patchwork as Butler's community—in his extravagant reading of biblical narratives.

His doctrine is an idiosyncratic interpretation of Christian traditions combined with a celebration of historical ecological figures (like Rachel Carson, or Saint Francis of Assisi) and the evocation of a companion species-oriented belief system, expressed in the songs of the group's Oral Hymnbook. As disaster approaches, his evocations of hope become more reluctant. Compared with most religious fashions currently on offer, Fredric Jameson writes, Edencliff, "with its prophets, its sermons, its taboos and even its Hymnbook, wears astonishingly well: ecological, communitarian, cunningly organised in decentralised units, each with its 'Ararat' of supplies stashed away against the inevitable Waterless Flood of plagues to come" [60]. Like most of Olamina's followers, Toby (one of the protagonists of *Year*) doubts the religious teachings of the elder generation but stays with the community because she shares their social code.

While religion functions as the emotional glue of a collective ideology, Jüliger's *Unfollow* concentrates on the messianic and apocalyptic motifs. His protagonist is obviously a Christ figure, a strange mix between Jesus and Mowgli. But he then turns into a satanic force, instigating his followers to commit attacks against mankind and bringing about its extirpation. Jüliger's offer of collective empowerment in service of a 'green' future is ultimately consumed by the figure of the ecoactivist as ecoterrorist. In addition, the human assassins are the executioners of a non-human agency—non-charismatic herbs, viruses, and fungi plotting to bring about the end of the Anthropocene and claiming total rule for themselves [45] (pp. 153–154). In recruiting naïve humans as suicide bombers, Earth acts as imperialistically as Al-Qaeda. Jüliger's biblically inspired but ultimately biocentric and antihumanist tale reiterates the ancient Western myth of nature is an evil force, eclipsing the romantic ecological view of nature as a benevolent caregiver more typical of YA fiction.

Dimaline and Brahmachari, writing from Indigenous and postcolonial perspectives, likewise rely on spiritual power as a binding force. While the Indigenous migrants in *Marrow Thieves* find emotional nourishment in the ancient stories transmitted by the elders, nature is itself inhabited by a comforting, protecting spirit guiding the children out of captivity in *River Runs Gold*. Expressing the deep emotional connectedness between humans and their fellow species, spirituality is a powerful agent of change.

## 10. Results

*Compost: Aesthetics of Repair*

Social-ecological transition stories suggest that the main forces in coping with global crisis are epistemic and spiritual resilience, cooperativeness and communication. The activity of rebuilding the world into a more livable place forms the main theme in many of the novels. With Butler's Parables as prototype, they combine an ethics of mutual aid with an aesthetics of repair. The Acorn community's school curriculum includes many practical skills: "We learn from all the work we do. We've become very competent makers and repairers of small tools. We've survived as well as we have because we keep learning" [41] (p. 26). The Parable novels teach the art of repair on various levels— materially, sociologically, psychologically. Butler's little group of ragged individuals, bound together by their various experiences of loss and trauma, discover hidden skills and practice the art of being content with a frugal life. This furnishes them with emotional resilience in situations of danger. They know that they can succeed only if working together as a group: "Here was real community. Here was at least a semblance of security. Here was the comfort of ritual and routine and the motional satisfaction of belonging to a 'team' that stood together to meet challenge when challenge came. . . . here was a place to raise children . . . " [41] (p. 61). Especially the YA novels insist on the primacy of generational justice, even while portraying intergenerational communication as fraught due to the parent generation's difficulties in coping with climate, ecological, and the resulting social crises. Yet Butler's portrayal of intergenerational conflict is politically unsatisfactory: Olamina's daughter, forcibly estranged from her mother and her project, later makes a living by inventing dreammasks, cheap "virtual-world fantasies" to sedate citizens' minds [41]

(p. 329). With a view to the magnitude of present real world fake fantasies, Butler's vision once again proved prescient.

At the present historical conjunction, Fredric Jameson famously wrote in 1994 (and again in 2003), it is easier for most people to imagine the end of the world rather than the end of capitalism [61] (p. xii), [62] (p. 76). Jameson's former student Kim Stanley Robinson suggests that an effective response to the climate challenge would have to include a systems change—and he accordingly wrote a systems change novel. His vision resembles Harvard economist Rebecca Henderson's who calculates that capitalism can be reformed into a more equitable, ecologically responsible system whose primary benefit is common wellbeing [63]. Donna Haraway proposes a different route—to disentangle the present from imagining "apocalyptic or salvific futures," proposing instead to "stay with the trouble as mortal critters entwined in myriad unfinished configurations of places, times, matters, meanings" [35] (p. 1). Resenting a "comic faith in technofixes"—a position Robinson largely shares—Haraway insists that what is required are "collaborations and combinations, in hot compost piles" [35] (p. 4). Confidence in humans' fitness for transformation is generated less by anticipations of a clean, high-tech-guided restart (Earth is too exhausted for that) than by "dwelling in the crisis" [29] (p. 246)—an epistemic change that would upcycle societies and the Earth itself from their (historical) "compost piles."

## 11. Conclusions: Disaster as Dwelling Place

This will have to include real compost piles. In the non-fiction field, a growing literature on traditional agricultural and ecological knowledge, which partly survived in non-Western rural societies and is partly reconstructed by practitioners, scholars and activists in metropolitan knowledge centers, suggests that an intelligent and critical assessment of past economic solutions—from a "tamed" market economy to sustainable agricultural, housing, and mobility practices—is a precondition of future living. Study of the (literature of the) past may reinvigorate knowledge of cultural strategies of repair that include both technical skills of fixing and mending (avoiding intentional obsolescence) and sociological techniques of recuperation, resilience, and recovery. Rebecca Solnit, who evokes historical cases of human solidarity and mutual help at times of crisis [58], calls for "changing the imagination of change" [64] (p. 60). The fictional texts discussed here contribute to that important project by describing patient, democratically conducted paths toward a less disastrous time ahead.

While the transition stories discussed here may help overcome the present paralysis in generating in readers a positive disposition toward behavioral change, in fact *any* story whose characters and actions exemplify sustainable behavior possesses this potential. Sara Pennypacker's YA novel *Here in the Real World* (2020) [65] is not set in the near future but, as the title states, here in the real world. This is a world in which soil is left to waste, land is treated as a speculation investment, and human spiritual needs are painfully neglected. Three children—a boy dreamer and two pretty realistic girls—try to save plants and animals by planting a vegetable garden in an urban waste plot, site of a demolished church and the annual resting area for migrating cranes. In the end the children have to give up their secret garden because the land is sold off by a real estate speculator and they remove their plants to the boy's parents' suburban garden—reminding readers of the hidden potential of the soil below the lawn wastes on their own properties. On a subliminal and perhaps not even intended level, the novel resemanticizes the historical origin of many of the contemporary world's problems: the early modern tragedy of the commons and dissolution of the monasteries. During this massive transformation of the rural economy since the late middle ages, commonly used farmland, especially in Great Britain, was forcefully enclosed with fences and transformed into large private estates while the farmers were forced to relocate to urban areas where they formed an urban proletariat. The process was ideologically rationalized by John Locke's theory of possessive individualism, an essential part of capitalist economic mythology. Many members of the deracinated rural population migrated to the colonies whose violent settlement policy is

a continuation of the original manmade "tragedy." Kim Stanley Robinson's dispossessed Montana village is an eerie echo of this historical process and its accompanying legend of voluntary migration—a reminder of the fact that the enclosure policy continues today unseen and in open daylight.

Pennypacker's story from the "real world" needs no imagined future for encouraging its readers to reflect on the possibility of making change in front of our doors, of experiencing disaster as a "dwelling place." The three children intensely reflect on how they may themselves contribute to the transition to a more ecological and climate-friendly land tenure, including the studious use of compost, even within the humble scope of urban backyard horticulture. They thereby imaginatively undo the enclosure practices of the past and present. The novel is a fine example of how literature can be an effective strategy in advancing the "great mindshift" Maja Göpel evokes [8].

Behavioral change depends, among many other things, on telling "sustainable" stories. Christiana Figueres and Tom Rivett-Carnac, the scientists who led the negotiations for the Paris Climate Agreement in 2015, explicitly invoke the "need" of literature and the arts to "become part of a new story of human striving and renewal" [3] (p. 158). Fictional and nonfictional narratives that generate confidence in mankind's ability of shaping an ecologically and socially sustainable future are an integral component of rebuilding the global economy. "It is through the imagination alone," Hannes Bergthaller writes, quoting John O'Grady, "that we come to recognize what 'truly sustains us'—our 'kinship' with the non-human world" [66] (p. 730). But romanticist appeals to the transformative power of the imagination and its "home turf," literature, Bergthaller argues, are not sufficient in and by themselves in establishing an ecologically sustainable world [66] (pp. 731, 735). As I noted above with reference to Nussbaum's thesis about the transformative power of literature, art and literature in general are no guarantees of an ecologically and socially responsible praxis. (Note: Suzanne Keen and Timothy Clark refer to the structural limits of narrative, especially the constraints of plot, for expressing complex problematics such as climate change, pointing out the comparable openness of poetry for contradiction and indeterminacy—although climate change poetry in Clark's view has not yet fully realized that potential [36] (p. 59). I agree with this position to the extent that conventional CliFi has been sadly uncreative in a utopian sense. This weakness of genre fiction responding to the expectations of a mass market, however, is not a weakness of narrative or the novel. The novel genre has been shown to possess a vast potential for imbricating various other textual forms, for polyvocality, contrapuntality, dialogism, and resistance to closure.) Discussing Atwood's God's Gardeners (in *Year*), Bergthaller writes that although their worldview "does not lay out a viable path to a sustainable future ... [,] their beliefs do in the end emerge as a credible response to the crisis" [66] (p. 738). Atwood's characters, too, have developed the skill of "dwelling" in disaster. Survival, Bergthaller summarizes Atwood's literary message in this novel, would fail unless accompanied by an "'anthropodicy'"—"a symbolic order within which the fact of survival can appear as meaningful and 'good'" [66] (p. 738). A truth easily to be missed because of the quaintness of the Gardeners' belief system is that it complements their subsistence farming with an "imaginary order that transcends" mere physical realities [66] (p. 739).

In imagining socially and ecologically responsible behavior, practical creativity, and the human capacity for frugality and reform, near future transition stories are among the "daydreams" necessary, according to Ernst Bloch (and correctly diagnosed by Cherie Dimaline), for moving beyond the precarious conditions of the present. They inspire the "concrete, knowledge-based hope" arising from "informed discontent" ("kundige Unzufriedenheit") about a situation of privation [67] (p. 3; my translation). "Hope," writes Rebecca Solnit in her book against despair, "is not a door, but a sense that there might be a door at some point, some way out of the problems of the present moment even before that way is found or followed" [64] (p. 22). Imagination is required for developing that sense of concrete, anticipatory hope. Social-ecological transition stories are part of mankind's cultural heritage for generating that hope.

Literature has always inspired human planning, whether technological innovation or innovative forms of governance. As Margaret Atwood and Ursula LeGuin state in the mottos above: Imagination is the precondition of action, and resistance and change often begin in literature. Near future transition narratives, rather than staying locked in delightful anxiety of global destruction, promote creative adaptation and change. Resilience, these writers know, springs from intellectual dispositions before finding expression in concrete social practices. The way the past, present, and future are integrated into continuous narratives of humans in constant interaction with the nonhuman world is important for thinking change and adaptation in a rapidly changing global environment. The dispositions conveyed in these transition stories, and performed by them on the readers, emphasize the need for individual courage while working cooperatively; the preservation of humanistic values while reaching out for the non-human; a readiness to confront novel situations while retaining the wisdom of ancestors; and the ability to fight fear and paralysis with the power of companionship, reason and science. Stories that transport these values and that portray the successful employment of the weapons of the weak in the face of antagonistic life-negating forces are fit for the future because they disseminate Gandhi's lesson about power's reaction to the powerless: "First they ignore you. Then they laugh at you. Then they fight you. Then you win" [64] (qtd. after Solnit, p. 61).

**Funding:** This research received no external funding.

**Institutional Review Board Statement:** Not applicable.

**Informed Consent Statement:** Not applicable.

**Data Availability Statement:** Not applicable.

**Conflicts of Interest:** The author declares no conflict of interest.

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
