# Peer review of "Sustainable Stories: Managing Climate Change with Literature†"

_sustainability, doi:10.3390/su13074049_

Round 1

Reviewer 1 Report

This is a very richt and strong paper with an inspiring argument for tranformative novels as more suitable for change than widespread dystopian science fiction and cli-fi. I would recommend to integrate the following recent publications which also explore the transformative and appealing power of cli-fi:

Goodbody Axel and Adeline Johns-Putra (eds.) (2019) Cli-Fi. A companion. Oxford et al.: Peter Lang.

Milner, Andrew and J.R. Burgmann (2020) Science Fiction and Climate Change: A Sociological Approach. Liverpool University Press, Liverpool.

Andersen, Gregers (2020) Climate Fiction and Cultural Analysis. A New Perspective on Life in the Anthropocene. New York: Routledge.

+ + +

A recommendation: In the paper, the term "transition stories/novels" might be more specified as "ecological transition stories" or "social-ecological transition novels".

I was wondering how transition stories/novels and sustainable texts are distinguished. Or is it by intention that these categories seem to blend and are used promiscuously?

Formal: 498 + 501: no line change

Author Response

Thanks for the helpful references, most of which I included.

Attached is the revised article, with track changes

Reviewer 2 Report

The article fits well within the field of ecocritical studies arguing for the effect of reading speculative fiction and other narrative fiction in promoting sustainable and/or ecocentric attitudes. The original and novel claim of the author is to present and discuss novels with positive outlooks on the future. I agree that this is an important and under-researched topic, although I would like to see more substantiation of the claim that "citizens' willingness to change elementary aspects of their lifestyle" (l. 64-65) is fundamental to reducing carbon emissions. Why does the author consider this an individual, rather than a governmental, responsibility?

The author bases her claims on theories of narrative ethics, in particular Martha Nussbaum. It would strengthen the argument if she also critically discussed the limitations of such theories. Suzanne Keen's Empathy and the Novel is an important reference. Furthermore, many ecocritics are growing skeptical of the claim that literature can transmit ecocentric values, e.g. Timothy Clark. I do not think the author should dodge this vexed discussion. At lines 856-858, she quotes Hannes Bergthaller to support a claim of the importance of imagination. The quote from Bergthaller, however, serves as a foil for his critique of what he terms "ecocritical orthodoxy," and thus this is a blatant misrepresentation of his work.

At times, the author makes some rather general and unsubstantiated claims about the relationship between capitalism, literary imagination, and the capacity of literature in raising ecological awareness. This also becomes somewhat repetitive, e.g. in lines 235-239, where the author reiterates her stance on the problems of "climateanxiety." My suggestion is to limit these assertions, in order to cut more directly to the chase: The analysis of various tropes and narrative structures characterizing literature likely to have empowering effects. This is the original and important knowledge contribution of the article, and should be highlighted.

In this regard, it would also be beneficial to quote some of the work that has been done on the potential of literature to empower readers, e.g. Janice Bland's work in the field of children's literature.

One minor comment: I would caution the author against misrepresenting the world we live in. However traumatic the Trump regime has been, it is simply not true that a depiction of an "authoritarian regime" where "autonomous communities are razed, their inhabitants incarcerated and their children abducted to be reeducated" is anything close to familiar to the United States of 2020 (lines 393-398).

Author Response

Thanks for the very helpful critique and further references. I included all changes in the attached revised article (track changes).

Reviewer 3 Report

comments are sent to the editor

Author Response

No response seemt to be requried to this review.